# Five-Year Trends in Potential Drug Interactions with Direct-Acting Oral Anticoagulants in Patients with Atrial Fibrillation: An Australian-Wide Study

**DOI:** 10.3390/jcm9113568

**Published:** 2020-11-05

**Authors:** Woldesellassie M. Bezabhe, Luke R. Bereznicki, Jan Radford, Barbara C. Wimmer, Mohammed S. Salahudeen, Ivan Bindoff, Edward Garrahy, Gregory M. Peterson

**Affiliations:** 1School of Pharmacy and Pharmacology, University of Tasmania, Private Bag 26, Hobart, Tasmania 7001, Australia; luke.bereznicki@utas.edu.au (L.R.B.); barbara.wimmer@utas.edu.au (B.C.W.); mohammed.salahudeen@utas.edu.au (M.S.S.); ivan.bindoff@utas.edu.au (I.B.); g.peterson@utas.edu.au (G.M.P.); 2Launceston Clinical School, Tasmanian School of Medicine, University of Tasmania, Launceston 7250, Australia; j.radford@utas.edu.au (J.R.); edward.garrahy@utas.edu.au (E.G.)

**Keywords:** atrial fibrillation, direct-acting oral anticoagulants, drug–drug interactions, primary care

## Abstract

Background: Co-prescribing medications that can interact with direct-acting oral anticoagulants (DOACs) may decrease their safety and efficacy. The aim of this study was to examine the co-prescribing of such medications with DOACs using the Australian national general practice dataset, MedicineInsight, over a five-year period. Methods: We performed five sequential cross-sectional analyses in patients with atrial fibrillation (AF) and a recorded DOAC prescription. Patients were defined as having a drug interaction if they had a recorded prescription of an interacting medication while they had had a recorded prescription of DOAC in the previous six months. The sample size for the cross-sectional analyses ranged from 5333 in 2014 to 19,196 in 2018. Results: The proportion of patients who had potential drug interactions with a DOAC decreased from 45.9% (95% confidence interval (CI) 44.6%–47.4%) in 2014 to 39.9% (95% CI 39.2%–40.6%) in 2018, *p* for trend < 0.001. During this period, the most frequent interacting class of medication recorded as having been prescribed with DOACs was selective serotonin/serotonin and norepinephrine reuptake inhibitor (SSRI/SNRI) antidepressants, followed by non-steroidal anti-inflammatory drugs (NSAIDs), calcium channel blockers (CCBs) and amiodarone. Conclusions: Overall, potential drug interactions with DOACs have decreased slightly over the last five years; however, the rate of possible interaction with SSRIs/SNRIs has remained relatively unchanged and warrants awareness-raising amongst prescribers.

## 1. Background

Atrial fibrillation (AF) is the most common sustained cardiac arrhythmia and occurs in about 2% of the general population [1,2]. Patients with AF are at a higher risk of developing stroke [3]. The newer direct-acting oral anticoagulants (DOACs) are being used more commonly to prevent ischaemic strokes and prolong life [4,5]. They are potentially safer, as effective and easier for administration compared with warfarin [6,7]. However, some medications have pharmacokinetic (PK) or pharmacodynamic (PD) interactions with DOACs and may decrease their safety and efficacy [8,9,10].

The absorption and metabolism of DOACs are highly dependent on P-glycoproteins (P-gp) and cytochrome P450 (CYP) 3A4 enzyme, respectively. The most important classes of drugs that alter the bioavailability and elimination of DOACs through inhibition of P-gp and CYP3A4 include macrolide antibiotics (e.g., erythromycin), azole antifungals (e.g., voriconazole) and HIV protease inhibitors (e.g., ritonavir) [8,11,12,13]. The subsequent rise in plasma levels of DOACs increases the risk of bleeding. Conversely, some antiepileptic medications, such as carbamazepine, decrease the plasma level of DOACs and their clinical effectiveness by inducing the metabolising enzyme, CYP3A4 [8,14].

In addition, there are PD interactions with DOACs. For instance, non-steroidal anti-inflammatory drugs (NSAIDs) [8,9], antiplatelet agents [8,15] and selective serotonin/serotonin and norepinephrine reuptake inhibitors (SSRIs/SNRIs) [8,16] increase the risk of bleeding when co-prescribed with DOACs, through pharmacological mechanisms.

There have been no large Australian studies investigating potential drug interactions in patients taking DOACs using real-world data. This study aimed to examine potential drug interactions in patients with AF taking DOACs over a five-year period.

## 2. Method

This study was an analysis of retrospective data obtained from the National Prescribing Service (NPS) MedicineWise dataset. The data were de-identified and extracted from the electronic health records (EHRs) of general practices across Australia. Details about this dataset can be found elsewhere [17,18,19].

### 2.1. Study Patients

We performed five sequential cross-sectional analyses at the end of each year from 2014 to 2018. Patients with a recorded diagnosis of AF and at least one DOAC prescription (dabigatran, rivaroxaban or apixaban) within the census year were included in the analysis. Patients aged <18 years at the end of the census year or who had had fewer than three visits to the same general practice within two years (including the census year and the year before the census) were excluded. Patients were defined as having a drug interaction in the census year if they had a recorded prescription of at least one interacting medication while they had had a recorded prescription of DOAC in the previous six months. We assumed a prescription of DOAC covered a maximum of six months.

The co-prescribed medications we searched for included NSAIDs (celecoxib, etoricoxib, lumiracoxib, parecoxib, rofecoxib, diclofenac, nepafenac, solifenacin, diflunisal, ibuprofen, indomethacin, ketoprofen, naproxen, sulindac, meloxicam, piroxicam, tenoxicam, ketorolac, mefenamic acid, flurbiprofen, tiaprofenic acid and phenylbutazone), antiplatelet agents (aspirin, clopidogrel, ticlopidine, prasugrel, ticagrelor, dipyridamole, abciximab, eptifibatide and tirofiban), macrolide antibiotics (erythromycin and clarithromycin), oral azole antifungals (ketoconazole, fluconazole, itraconazole, voriconazole, and posaconazole), HIV protease inhibitors (amprenavir, atazanavir, darunavir, fosamprenavir, indinavir, lopinavir, ritonavir, nelfinavir and saquinavir), calcium channel blockers (CCBs) (diltiazem and verapamil), antiarrhythmic agents (amiodarone), antiepileptics (phenytoin, carbamazepine, valproate, levetiracetam and primidone) and selective serotonin/norepinephrine reuptake inhibitors (SSRIs/SNRIs) (citalopram, escitalopram, duloxetine, fluoxetine, paroxetine, fluvoxamine, sertraline, nefazodone, desvenlafaxine and venlafaxine) [11,20].

We obtained ethics approval for this study from the University of Tasmania’s Human Research Ethics Committee (H0017648). An approval to conduct this study was also obtained from the MedicineInsight independent Data Governance Committee (2018-033).

### 2.2. Outcome

The primary outcome of interest was the proportion of patients with a recorded prescription of DOACs with potential interacting medications.

### 2.3. Statistical Analysis

The proportion of patients with potential drug interactions was calculated at the end of each year from 2014 to 2018. We used a Cochran–Armitage test for trend to determine if any observed trends were statistically significant. All analyses were performed using SAS software (SAS version 9.4, SAS Institute Inc., Cary, NC, USA). A two-sided *p*-value < 0.001 was considered statistically significant.

## 3. Results

The total number of patients included in each census year ranged from 5333 in 2014 to 19,196 in 2018. The mean age of the 2018 cohort was 76.7 years and 10,410 (54.2%) were males. Overall, the proportion of patients who had a potential drug interaction with DOAC decreased from 45.9% (95% confidence interval (CI) 44.6%–47.4%) in 2014 to 39.9% (95% CI 39.2%–40.6%) in 2018, *p* for trend <0.001. During this period, the most frequent interacting class of medication prescribed with DOAC was SSRIs/SNRIs, followed by NSAIDs, CCBs and amiodarone (Figure 1 and Appendix A). The proportion of patients who had a CCB interaction with DOACs decreased from 11.3% (95% CI 10.4%–12.1%) in 2014 to 8.8% (95% CI 8.4%–9.2%) in 2018. The proportion who had an NSAID interaction over the same period also decreased slightly, from 11.6% (95% CI 10.7%–12.4%) to 9.7% (95% CI 9.3%–10.1%). The rate of interaction with SSRIs/SNRIs or amiodarone relatively remained unchanged at approximately 14% and 6%, respectively, over the five-year period.

In terms of individual DOACs, 40.0% (1105/2766), 39.1% (3057/7816) and 40.5% (3491/8614) of patients taking dabigatran, apixaban and rivaroxaban, respectively, were prescribed at least one concurrent potentially interacting medication in 2018 (Table 1).

## 4. Discussion

This nationwide general practice study shows the following main results. Firstly, some specific medications advised to be avoided in DOAC users [8,11,21], including SSRIs/SNRIs, NSAIDs and amiodarone, were commonly prescribed to patients with AF. During 2018, for instance, SSRIs/SNRIs, NSAIDs and CCBs were co-prescribed with DOACs in 14.8%, 9.7% and 8.8% of DOAC users, respectively. Secondly, perhaps due to increasing awareness, the rate of potential drug interactions with DOACs showed a decreasing trend over a five-year period; however, the rate of prescribing SSRIs/SNRIs was relatively high and remained unchanged. Similarly, a high rate of concomitant SSRIs/SNRI and DOAC prescribing (22.9%, 22/122) was also reported in a small Australian study of elderly hospitalised patients by Forbes et al. [22]. Notably, a recent study by Zhang et al. [16], involving almost 24,000 new users of DOACs in the UK Clinical Practice Research Datalink, found a significant increase in the risk of major bleeding in patients co-prescribed an SSRI/SNRI (adjusted odds ratio (aOR), 1.68 (95% confidence interval [CI], 1.10–2.59)) but not with co-prescribed PK-interacting drugs, such as amiodarone. A study by Quinin et al. [23] also reported that major bleeding was higher in patients taking warfarin during periods of SSRI exposure compared with periods with warfarin alone (2.32 per 100 person-years, *p* < 0.001).

One of the possible explanations for the high proportion of patients co-prescribed an SSRI/SNRI is a lack of awareness on the interaction of SSRIs/SNRIs with DOAC. There may be a need to raise prescribers’ awareness of this interaction. The incidence of drug interactions was lower than in a Taiwanese study [24] by Chang et al. [24], in which diltiazem and amiodarone were used in more than 20% of DOAC-exposed person-quarters (exposure time for each person during each quarter of the calendar year). A possible explanation for this difference might be our use of more recent data, as compared with 2012–2016 data for the Taiwanese study.

### Strengths and Limitations

This is the first Australian study that evaluated potential drug interactions in patients prescribed DOACs, using the largest general practice dataset available to researchers. Busingye et al. [25] investigated the national representativeness of the NPS MedicineWise dataset. While South Australia was underrepresented and Tasmania and New South Wales were overrepresented, the dataset was representative in terms of the age and gender of the participants based on comparisons with national Medical Benefits Schedule data. Although we only included regular general practice patients who had at least three visits within two years in each cross-sectional analysis, it is possible that interacting medicines were prescribed by other general practices or specialists and not recorded in the MedicineInsight dataset. The use of aspirin or NSAIDs is not routinely recorded in the dataset, as many patients obtain them without a prescription. Patients were defined as having a drug interaction if they had a recorded prescription of an interacting medication while they had had a recorded prescription of DOAC in the previous six months. As a result, it is possible that the DOAC had been ceased prior to the actual commencement of the interacting drug.

This study did not have clinical outcome data, such as bleeding and ischaemic stroke, associated with the potential drug interactions. However, elsewhere, patients who were prescribed the selected interacting medications for this study with DOACs were found to experience major bleeding or ischaemic stroke [14,16,24]. It should be noted that the rates alone of co-prescribing of potentially interacting drugs do not necessarily correlate with the risk of major adverse events. For instance, although the rate of co-prescribing macrolides and azole antifungals with DOACs was low (i.e., 2.9% and 0.7%, respectively, for 2018), the risk for clinically severe consequences could have been high, based on their PK interactions. Similarly, the duration of co-prescription of interacting medications would influence the risk of significant adverse events arising. For instance, SSRIs/SNRIs are co-prescribed with DOACs mostly for long-term conditions, and this may partly explain the observed increased risk of bleeding with these drugs. It might also partly explain why the co-prescribing of SSRIs/SNRIs remained relatively constant, in comparison with drugs more commonly used short-term (e.g., NSAIDs) over the course of the study.

## 5. Conclusions

Overall, potential drug interactions with DOACs have decreased slightly over the last five years; however, the rate of possible interaction with SSRIs/SNRIs has remained relatively unchanged and may warrant awareness-raising amongst prescribers.

## Figures and Tables

**Figure 1 jcm-09-03568-f001:**
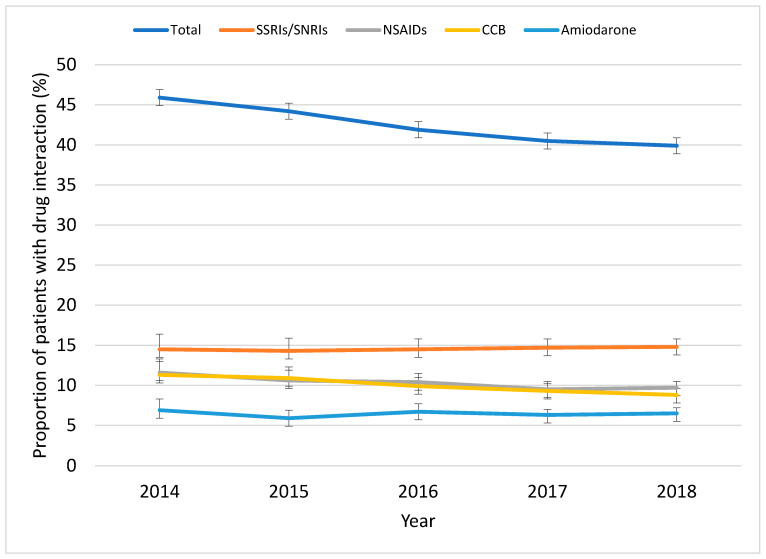
Trends in potential drug interactions in patients taking direct-acting oral anticoagulants.

**Table 1 jcm-09-03568-t001:** Concomitant medications prescribed with direct-acting oral anticoagulants in 2018.

Concomitant Medications	Total (*n* = 19,196)	Dabigatran (*n* = 2766)	Rivaroxaban (*n* = 7816)	Apixaban (*n* = 8614)
All interacting medications (%)	7653 (39.9)	1105 (40.0)	3057 (39.1)	3491 (40.5)
Calcium channel blockers (%)	1687 (8.8)	263 (9.5)	684 (8.8)	740 (8.6)
Amiodarone (%)	1250 (6.5)	629 (7.3)	472 (6.0)	629 (7.3)
Azole antifungals (%)	137 (0.7)	18 (0.7)	63 (0.8)	56 (0.7)
Macrolides (%)	548 (2.9)	84 (3.0)	228 (2.9)	236 (2.7)
Antiepileptics (%)	409 (2.1)	57(2.1)	151 (1.9)	201 (2.3)
NSAIDs (%)	1864 (9.7)	275 (9.9)	807 (10.3)	782 (9.1)
Antiplatelet agents (%)	968 (5.0)	145 (5.2)	324 (4.1)	499 (5.8)
SSRIs/SNRIs (%)	2849 (14.8)	408 (14.8)	1141 (14.6)	1300 (15.1)

NSAIDs, non-steroidal anti-inflammatory drugs; SNRIs, selective serotonin and norepinephrine reuptake inhibitors; SSRIs, selective serotonin reuptake inhibitors.

## Data Availability

The data owner, NPS MedicineWise, restricts data sharing. Thus, data for this study are not available for the public.

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
