# Peer review of "Five-Year Trends in Potential Drug Interactions with Direct-Acting Oral Anticoagulants in Patients with Atrial Fibrillation: An Australian-Wide Study"

_jcm, 2020, doi:10.3390/jcm9113568_

Round 1

Reviewer 1 Report

In this retrospective study Bezabhe et al. describe co-prescribing of interactive medications with DOACs using the Australian national dataset, MedicineInsight.  

The paper is clearly written.

The number of cited articles is suitable and covers quite well the appropriate literature.

My remarks are:

The first references describing epidemiology are outdated. The fresher would be for example Staerk, Circ Res. 2017;120:1501-1517, or Staerk, BMJ 2018;360:k1453 and Norberg Clin Epidemiol. 2013;5:475-81.

MedicineInsight practices cover about 8% of national practices in Australia. Is this a representative sample of Australian people?

Could the result have been different, if MedicineInsight has wider coverage?  

Have there been changes of characteristics of patients on NOACs during the study period?

Now the changes of the use of interactive medications during the five-year study period were very modest.

The number of patients on NOACs has remarkably increased, but has the patients been alike?

The references should be interpreted in one and the same way. With or without first names.

Author Response

Response to Reviewer 1 Comments

Point 1: The first references describing epidemiology are outdated. The fresher would be for example Staerk, Circ Res. 2017;120:1501-1517, or Staerk, BMJ 2018;360:k1453 and Norberg Clin Epidemiol. 2013;5:475-81.

Response 1: Thank you for your comment. We have now cited Staerk et al. (2018), Zulkifly et al. (2018), and Benjamin et al. (2019) instead of the old references.

Point 2: MedicineInsight practices cover about 8% of national practices in Australia. Is this a representative sample of Australian people? Could the result have been different, if MedicineInsight has wider coverage?  

Response 2: Thanks for asking this; we have included the following statement in the “Strengths and limitations” section. “Busingye et al.[24] investigated the national representativeness of the NPS MedicineWise dataset. While South Australia was underrepresented and Tasmania and New South Wales were overrepresented, the dataset was representative in terms of age and gender of the participants based on comparisons with national Medical Benefits Schedule data.” Given these characteristics, we believe that it is reasonable to assume that the results of this analysis would not be different if MedicineInsight had wider coverage.

Point 3: Have there been changes of characteristics of patients on NOACs during the study period?

Now the changes of the use of interactive medications during the five-year study period were very modest.

The number of patients on NOACs has remarkably increased, but has the patients been alike?

Response 3:

Patients were included in each cross-sectional analysis if they were aged 18 years or older, had AF and met the Royal Australian College of General Practice (RACGP) definition of a regular patient. A regular patient was defined as a patient who had at least three consultations with a general practitioner at the same general practice within a two-year period. However, we did not specifically analyse the data to see changes in the characteristics of the patients prescribed DOACs. We feel that this would sit better in a separate project.  

Point 4: The references should be interpreted in one and the same way. With or without first names.

Response 4:

Thanks for your comments. We noted this in our discussion section, and we now consistently refer to references using the authors’ last name.

Reviewer 2 Report

In this study, the co-prescription of a DOAC (dabigatran, rivaroxaban or apixaban) for non-valvular atrial fibrillation (AF), and at least one drug with established potential interaction was recorded from a large national database in Australia. The data were analysed by year between 2014 and 2018. The study provides interesting results. First, the co-prescription of DAOC and any interacting drug decreases significantly but very slightly with time (about 1 % per year) whereas the total prescription of a DOAC increased gradually in a much larger proportion each year (+55 % the 1st year, + 40 % the 2nd, etc.). Second, the most frequent interacting drugs are SSRIs/SNRIs (this may reflect the large use of this class of drug in the general population of aged patients), and this proportion is remarkably stable (14.5-14.9 %).

I have 2 major comments related to the methodology:

The actual time spent on co-prescription is not indicated but it is likely that SSRIs/SNRIs treatment is on long term, if not definitive. In contrast, NSAID is often prescribed for a short period (e.g.: one week). Intuitively, the risk of adverse effect (bleeding) is in proportion of time. If so, co-prescription of SSRIs/SNRIs should lead to more events compared to NSAIDs. On the other hand, the level of interaction is not uniform between the classes of drugs. The co-prescription of a potent interacting drug such as macrolides or azole antifungals, even for 1-2 weeks, may be more dangerous than SSRIs/SNRIs with less strong interaction. Therefore, the risk is not proportional to the simple proportion of patients, but depend mainly on the type of drug and the length of co-prescription.

Globally, the results suggest that in spite of continuous information of the prescribers and the patients on the good practice with DOACs, there is no real improvement. However, DOACs are prescribed definitely in the majority of patients with AF. That means that among the 5534 patients taking a DOAC in 2014, a large proportion of them (with or without a co-prescription) is counted in the following years, idem for the new patients in 2015, etc. This could explain in large part the stability of the results for SRIs/SNRIs (long term-treatment), whereas NSAID (usually short-term) and CCB (possible alternative) co-prescriptions slightly decrease. Alternatively, the effect of SRIs/SNRIs is less known by the prescribers, compared to other drugs.

Could the authors discuss these points?

Author Response

Response to Reviewer 2 Comments

In this study, the co-prescription of a DOAC (dabigatran, rivaroxaban or apixaban) for non-valvular atrial fibrillation (AF), and at least one drug with established potential interaction was recorded from a large national database in Australia. The data were analysed by year between 2014 and 2018. The study provides interesting results. First, the co-prescription of DAOC and any interacting drug decreases significantly but very slightly with time (about 1 % per year) whereas the total prescription of a DOAC increased gradually in a much larger proportion each year (+55 % the 1st year, + 40 % the 2nd, etc.). Second, the most frequent interacting drugs are SSRIs/SNRIs (this may reflect the large use of this class of drug in the general population of aged patients), and this proportion is remarkably stable (14.5-14.9 %).

I have 2 major comments related to the methodology:

Point 1: The actual time spent on co-prescription is not indicated but it is likely that SSRIs/SNRIs treatment is on long-term, if not definitive. In contrast, NSAID is often prescribed for a short period (e.g.: one week). Intuitively, the risk of adverse effect (bleeding) is in the proportion of the time. If so, co-prescription of SSRIs/SNRIs should lead to more events compared to NSAIDs. On the other hand, the level of interaction is not uniform between the classes of drugs. The co-prescription of a potent interacting drug such as macrolides or azole antifungals, even for 1-2 weeks, maybe more dangerous than SSRIs/SNRIs with less strong interaction. Therefore, the risk is not proportional to the simple proportion of patients but depend mainly on the type of drug and the length of co-prescription.

Response 1:

Thanks for this suggestion. We agree that the risk of an adverse event related to a drug interaction depends on a number of factors, including the types of co-prescribed medication and the duration of co-prescription. We have now elaborated on this theme in the discussion: “This study did not have clinical outcomes data, such as bleeding and ischaemic stroke, associated with the potential drug interactions. However, elsewhere, patients who have prescribed the selected interacting medications for this study with DOACs were found to experience major bleeding or ischaemic stroke [19, 29, 16]. It should be noted that the rates, alone, of co-prescribing of potentially interacting drugs, do not necessarily correlate with the risk of major adverse events. For instance, although the rate of co-prescribing of macrolides and azole antifungals with DOACs was low (e.g. 2.9% and 0.7%, respectively, for 2018), the risk for clinically severe consequences could have been high based on their PK interactions. Similarly, the duration of co-prescription of interacting medications would influence the risk of significant adverse events arising. For instance, SSRIs/SNRIs are co-prescribed with DOACs mostly for long-term conditions, and this may partly explain the observed increased risk of bleeding with these drugs. It might also partly explain why the co-prescribing of SSRIs/SNRIs remained relatively constant, in comparison with drugs more commonly used short-term (e.g. NSAIDs), over the course of the study.”

Point 2: Globally, the results suggest that in spite of continuous information of the prescribers and the patients on the good practice with DOACs, there is no real improvement. However, DOACs are prescribed definitely in the majority of patients with AF. That means that among the 5534 patients taking a DOAC in 2014, a large proportion of them (with or without a co-prescription) is counted in the following years, idem for the new patients in 2015, etc. This could explain in large part the stability of the results for SRIs/SNRIs (long term-treatment), whereas NSAID (usually short-term) and CCB (possible alternative) co-prescriptions slightly decrease. Alternatively, the effect of SRIs/SNRIs is less known by the prescribers, compared to other drugs.

Could the authors discuss these points?

Response 2:

Thanks for your suggestion. This has now been added to the same discussion paragraph as above: “This study did not have clinical outcomes data, such as bleeding and ischaemic stroke, associated with the potential drug interactions. However, elsewhere, patients who have prescribed the selected interacting medications for this study with DOACs were found to experience major bleeding or ischaemic stroke [19, 29, 16]. It should be noted that the rates, alone, of co-prescribing of potentially interacting drugs, do not necessarily correlate with the risk of major adverse events. For instance, although the rate of co-prescribing of macrolides and azole antifungals with DOACs was low (e.g. 2.9% and 0.7%, respectively, for 2018), the risk for clinically severe consequences could have been high based on their PK interactions. Similarly, the duration of co-prescription of interacting medications would influence the risk of significant adverse events arising. For instance, SSRIs/SNRIs are co-prescribed with DOACs mostly for long-term conditions, and this may partly explain the observed increased risk of bleeding with these drugs. It might also partly explain why the co-prescribing of SSRIs/SNRIs remained relatively constant, in comparison with drugs more commonly used short-term (e.g. NSAIDs), over the course of the study.”